# ERG signaling in prostate cancer is driven through PRMT5-dependent methylation of the Androgen Receptor

Zineb Mounir[1†], Joshua M Korn[1], Thomas Westerling[2,3,4], Fallon Lin[1], Christina A Kirby[5], Markus Schirle[6], Gregg McAllister[6], Greg Hoffman[6], Nadire Ramadan[6], Anke Hartung[7‡], Yan Feng[6], David Randal Kipp[1§], Christopher Quinn[1§], Michelle Fodor[1§], Jason Baird[1§], Marie Schoumacher[1§], Ronald Meyer[1], James Deeds[1], Gilles Buchwalter[2,3,4¶], Travis Stams[5], Nicholas Keen[1], William R Sellers[1], Myles Brown[2,3,4], Raymond A Pagliarini[1*]

[1]Department of Oncology, Novartis Institutes for BioMedical Research, Cambridge, United States; [2]Department of Medical Oncology, Harvard Medical School, Boston, United States; [3]Center for Functional Cancer Epigenetics, Harvard Medical School, Boston, United States; [4]Dana-Farber Cancer Institute, Harvard Medical School, Boston, United States; [5]Center for Proteomic Chemistry, Novartis Institutes for BioMedical Research, Cambridge, United States; [6]Developmental and Molecular Pathways, Novartis Institutes for Biomedical Research, Cambridge, United States; [7]Genomics Institute of the Novartis Research Foundation, Novartis Institutes for Bio Medical Resarch, San Diego, United States

*For correspondence: raymond.pagliarini@novartis.com

Present address: [†]Genentech, South San Francisco, United States; [‡]Organovo, San Diego, United States; [§]Laboratories Servier, Neuilly-sur-Seine, France; [¶]Celgene Avilomics Research, Bedford, United States

**Abstract** The *TMPRSS2:ERG* gene fusion is common in androgen receptor (AR) positive prostate cancers, yet its function remains poorly understood. From a screen for functionally relevant ERG interactors, we identify the arginine methyltransferase PRMT5. ERG recruits PRMT5 to AR-target genes, where PRMT5 methylates AR on arginine 761. This attenuates AR recruitment and transcription of genes expressed in differentiated prostate epithelium. The AR-inhibitory function of PRMT5 is restricted to *TMPRSS2:ERG*-positive prostate cancer cells. Mutation of this methylation site on AR results in a transcriptionally hyperactive AR, suggesting that the proliferative effects of ERG and PRMT5 are mediated through attenuating AR's ability to induce genes normally involved in lineage differentiation. This provides a rationale for targeting PRMT5 in *TMPRSS2:ERG* positive prostate cancers. Moreover, methylation of AR at arginine 761 highlights a mechanism for how the ERG oncogene may coax AR towards inducing proliferation versus differentiation.

## Introduction

Prostate cancer (PC) is highly prevalent and lethal (*Siegel et al., 2015*). Drugs targeting the Androgen Receptor (AR), a 'lineage driver' of PC (*Garraway and Sellers, 2006*), are an important therapeutic approach. AR is an androgen (i.e. testosterone)-activated nuclear hormone receptor that regulates normal prostate gland growth and differentiation. In PC however, AR facilitates unregulated proliferation (*Mills, 2014*). While it is unclear how AR and other lineage factors switch between promoting normal lineage differentiation vs. tumor growth, it is hypothesized that somatic mutations in additional genes may facilitate such changes (*Garraway and Sellers, 2006*). Many PCs bear chromosomal translocations resulting in aberrant expression of the ETS transcription factor ERG, most commonly through the *TMPRSS2:ERG* fusion (*Shah and Chinnaiyan, 2009*). *TMPRSS2:ERG*, alone or

**eLife digest** Prostate cancers are among the most common types of cancer in men, which, like other cancers, are driven by genetic mutations. Roughly half of all prostate cancers contain a genetic change that incorrectly fuses two genes together, causing the cells to produce abnormally high levels of a protein called ERG.

ERG is a transcription factor, a protein that binds to specific sequences of DNA to influence the activity of nearby genes. ERG represses genes that help to prevent prostate cancers from growing, and so promotes prostate cancer development. Like most other transcription factors, ERG is difficult to target with drugs and no therapies that directly prevent the activity of ERG currently exist.

Mounir et al. wanted to find out whether ERG cooperates with other proteins to cause prostate cancer cells to grow, with the hope that these proteins could be more easily targeted with a drug. By using various biochemical techniques in human prostate cancer cell lines, Mounir et al. found that ERG interacts with an enzyme called PRMT5. This interaction enables PRMT5 to chemically modify other proteins to change their activity. In the case of prostate cancer cells, PRMT5 inappropriately modifies the androgen receptor, a protein that regulates the growth of normal prostate cells. This abnormal modification contributes to the excessive growth of the cancer cells.

Although PRMT5 will be easier to target with drugs than ERG, it also has many other roles besides those described by Mounir et al. Much more work is therefore needed to investigate whether PRMT5 could be safely targeted to treat patients with prostate cancer.

in combination with additional genetic alterations, promotes prostate tumor formation in mice (*Baena et al., 2013*; *Chen et al., 2013*; *King et al., 2009*; *Klezovitch et al., 2008*; *Mounir et al., 2015*; *Tomlins et al., 2008*). ERG is recruited to many AR target genes and represses AR-dependent transcription (*Yu et al., 2010*), suggesting ERG functions at least in part through attenuating AR target gene expression. However, ERG also regulates the expression of AR-independent genes thought to drive oncogenic function (*Klezovitch et al., 2008*; *Mounir et al., 2015*; *Tomlins et al., 2008*; *Wang et al., 2008*). We explored whether a deeper mechanistic understanding of ERG proliferative function could yield therapeutic insights into targeting this key PC oncogene.

## Results and discussion

To identify genes that selectively facilitate the growth of *TMPRSS2:ERG* positive PC cells, we performed a pooled short hairpin RNA (shRNA) screen in *TMPRSS2:ERG* and AR-positive VCaP prostate cancer cells, using ERG-negative 22Rv1 cells as a control (Materials and methods). The shRNA pool targets 648 genes involved in transcriptional and epigenetic regulation (*Supplementary file 1*). While ERG shRNAs were not in the pool, AR shRNAs were preferentially depleted from VCaP cells, underscoring AR dependence in this cell line. Thirty two (32) genes showing VCaP-selective shRNA depletion (Materials and methods) were considered for further study (*Figure 1A*; *Supplementary file 1*).

We next narrowed the shRNA screen hit list by focusing on candidates more likely to be ERG interacting proteins. We immunoprecipitated ERG from VCaP cells, and then identified co-immunoprecipitated proteins by mass spectrometry (Materials and methods). Identified proteins (*Supplementary file 2*) included AR and DNA-PKcs, previously known ERG interactors (*Brenner et al., 2011*; *Yu et al., 2010*). Eight of the VCaP-selective shRNA screen hits that also co-immunoprecipitated with ERG were further validated by directed ERG co-immunoprecipitation experiments in VCaP cells. Of these, AR and PRMT5 were the only proteins that co-immunoprecipitated with ERG but not IgG control; these interactions were not overtly influenced by exposure to an androgen analog (R1881, *Figure 1B*; *Figure 1—figure supplement 1A*). We next tested whether the ERG/PRMT5 interaction is observed in other models. PRMT5 co-immunoprecipitated with ERG in 22Rv1 cells ectopically expressing ERG. This interaction was still observed upon expression of ERG bearing mutations in the DNA binding domain ('Dx', *Figure 1C*), suggesting DNA binding is not required for the ERG/PRMT5 interaction. Reciprocal co-immunoprecipitation experiments using overexpressed ERG and PRMT5 in AR-negative 293 and PC3 cells suggest the ERG/PRMT5

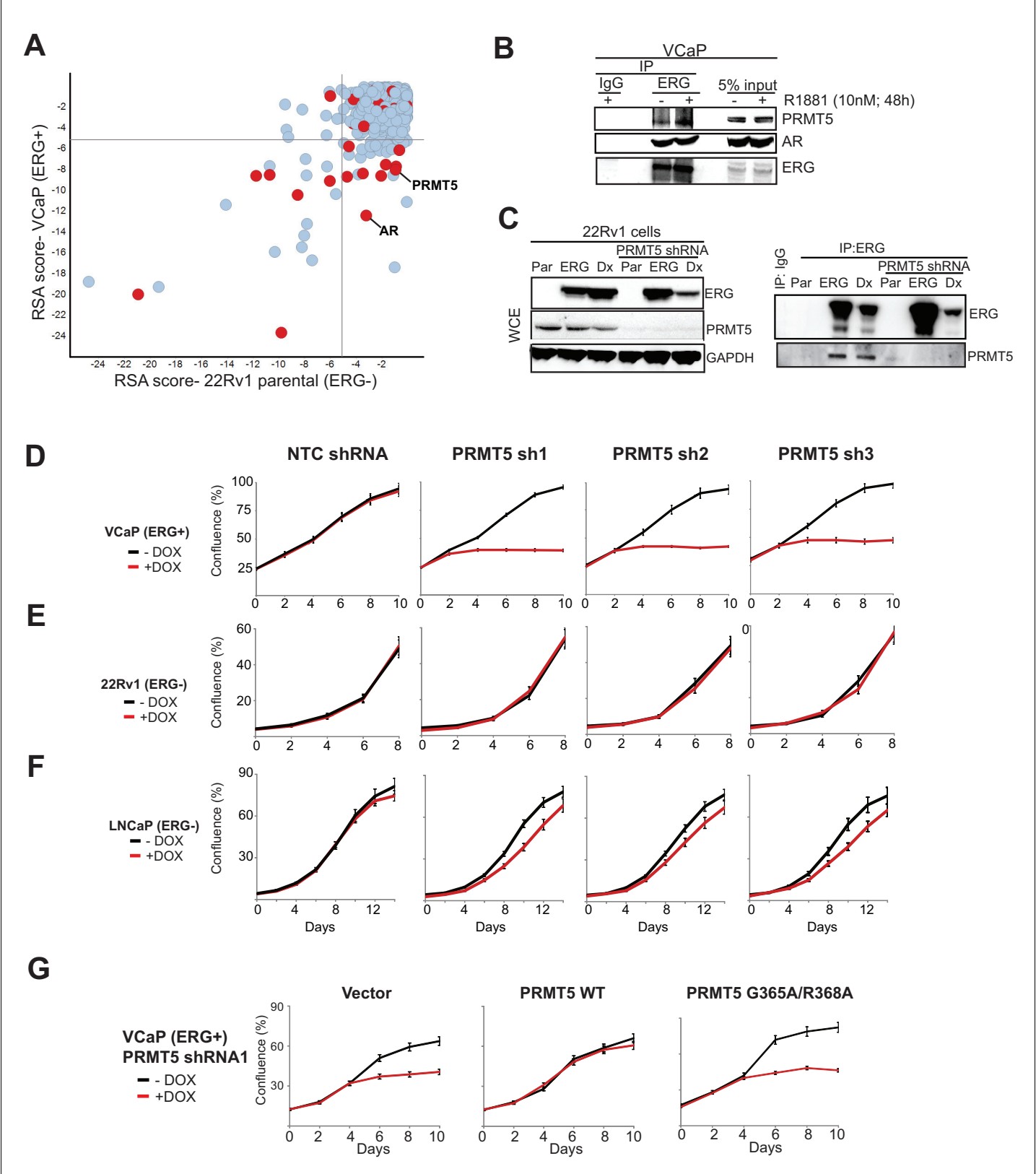

**Figure 1.** Identification of PRMT5. (**A**) Log p-value plots (RSA metric, see Materials and methods) of shRNA depletion from VCaP cells (y-axis) versus 22Rv1 cells (x-axis). Grey lines denote *p*-value cutoff for screen hits (10^-5), with bottom right quadrant enriched for VCaP-selective screen hits. Red dots indicate screen hits that are also candidate ERG interactors from ***Supplementary file 2.*** (**B**) Western blot of PRMT5, AR, and ERG following ERG or
*Figure 1 continued on next page*

*Figure 1 continued*

control IgG immunoprecipitation from untreated (-) or R1881-treated (+) VCaP cells. (C) Left panel: western blots of noted proteins from 22Rv1 whole cell extracts (WCE), either in parental (Par) cells, cells expressing exogenous ERG ('ERG'), or cells expressing a DNA-binding defective ERG ('Dx'); PRMT5 knockdown under these conditions is as noted. Right panel: Western blot of ERG immunoprecipitation (IP) from 22Rv1 for ERG and PRMT5. (D) PRMT5 proliferation after PRMT5 knockdown (sh1, sh2 and sh3) in VCaP cells (see Materials and methods). NTC: non-targeting control. Error bars represent ± SEM of three biological replicates, each with three technical repeats. (E) 22Rv1 proliferation as in (C). (F) LNCaP proliferation as in (C). (G) VCaP proliferation as in (C) alongside expression of shRNA-resistant wild-type (WT) PRMT5, catalytically inactive PRMT5 (G365A/R368A), or vector control (Vector). Error bars represent ± SEM of three biological replicates, each with three technical repeats.

The following figure supplement is available for figure 1:

**Figure supplement 1.** PRMT5 knockdown in prostate cancer cells.

interaction can occur in the absence of AR (*Figure 1—figure supplement 1B*). Further work in 293 cells using truncated ERG constructs suggested that the conserved ETS DNA binding domain of ERG was necessary for the observed co-immunoprecipitation with PRMT5 (*Figure 1—figure supplement 1C*). Given this evidence that ERG and PRMT5 co-exist in a protein complex, we focused further efforts on PRMT5, as to our knowledge it has not been previously linked to ERG biology.

To validate the growth effects of PRMT5 knockdown, we transduced ERG-positive VCaP cells, and ERG-negative 22Rv1 and LNCaP PC cells, with three independent doxycycline (Dox)-inducible shRNA vectors targeting PRMT5 and a non-targeting control shRNA (NTC). PRMT5 knockdown was robust in all cell lines (*Figure 1—figure supplement 1D*). Robust growth inhibition was observed in VCaP cells; in contrast PRMT5 knockdown had no growth effects in ERG-negative 22Rv1 cells, and only minor effects in ERG negative LNCaP cells (*Figure 1D–F*). Deletion of methylthioadenosine phosphorylase (MTAP), which is common across cancers, is a major determinant of sensitivity to PRMT5 inhibition (*Kryukov et al., 2016*; *Mavrakis et al., 2016*); as VCaP, LNCaP, and 22Rv1 cells are all MTAP intact, the observed sensitivity of VCaP to PRMT5 shRNA is not due to MTAP deletion. The project Achilles shRNA screen dataset (*Kryukov et al., 2016*) contains three prostate cancer cell lines (VCaP, 22Rv1 and *TMPRSS2:ERG* positive NCI-H660) and one PRMT5 hairpin likely to have minimal off-target effects. This shRNA shows a trend of sensitivity in ERG-positive lines, in agreement with our findings (*Figure 1—figure supplement 1E*).

PRMT5 is a protein arginine methyltransferase that regulates multiple signaling pathways through the mono- and symmetric di-methylation of arginines on its target proteins (*Yang and Bedford, 2013*).To determine whether the antiproliferative effects of PRMT5 knockdown in ERG positive VCaP cells were mediated through methyltransferase activity, we expressed shRNA-resistant wild-type PRMT5, or a catalytically inactive G365A/R368A double mutant (Materials and methods) (*Antonysamy et al., 2012*) along with PRMT5 shRNA in VCaP cells. WT PRMT5, but not the G365A/R368A mutant, rescued the effects of PRMT5 knockdown on VCaP cell proliferation (*Figure 1G*), indicating a requirement for PRMT5 catalytic function to support VCaP proliferation.

To understand pathways affected by PRMT5, we performed transcriptional profiling of PRMT5 knockdown in VCaP cells, followed by the identification of significantly altered pathways (*Figure 2A*; *Supplementary file 3*; see Materials and methods). Among these, AR activation was the second most significantly affected pathway, and is a key pathway in common with previous reports of ERG knockdown in VCaP cells (*Chen et al., 2013*; *Mounir et al., 2015*; *Yu et al., 2010*). AR pathway upregulation was apparent using multiple published AR gene signatures (*Figure 2—figure supplement 1*). Using quantitative PCR of reverse transcribed RNA (qRT-PCR), we confirmed that knockdown of either PRMT5 or ERG increased the expression of the AR target genes *PSA*, *NKX3-1* and *SLC45A3* (*Figure 2B*). Expression of shRNA-resistant WT PRMT5, but not the G365A/R368A mutant, rescued the effects of PRMT5 knockdown on AR target gene expression (*Figure 2C*; *Figure 2—figure supplement 2A*), demonstrating that PRMT5 methyltransferase activity is required for repression of AR target genes. The effect of PRMT5 knockdown was restricted to genes co-regulated by both AR and ERG, as PRMT5 knockdown did not affect previously published (*Mounir et al., 2015*) AR-independent ERG target genes in VCaP (*Figure 2—figure supplement 2B*). In addition, PRMT5 knockdown did not induce AR target gene expression in ERG negative 22Rv1 or LNCaP PC cells (*Figure 2—figure supplement 2C*). In 22Rv1 cells, exogenous ERG expression is sufficient to

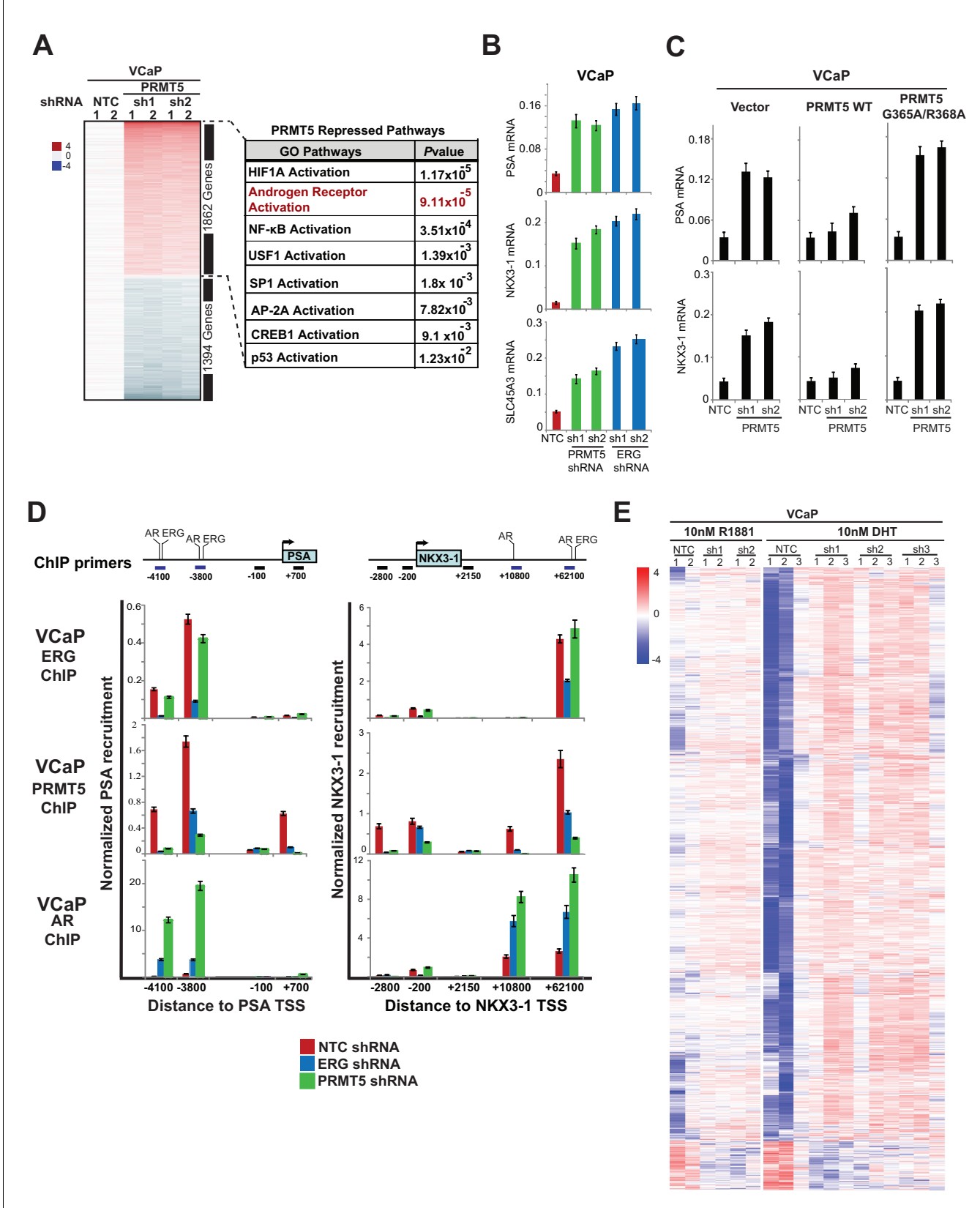

**Figure 2.** PRMT5 is an ERG-dependent inhibitor of AR signaling. (**A**) Heat map showing all genes upregulated (red) or downregulated (blue) by at least 1.5 fold following knockdown with PRMT5 shRNA1 (sh1) or shRNA2 (sh2) compared to NTC shRNA. Rows represent probe sets; columns represent

*Figure 2 continued*

individual samples (technical replicates are marked by 1 or 2). Table indicates pathways significantly upregulated by PRMT5 knockdown (see Materials and methods, and *Supplementary file 3* for significantly downregulated pathways). (B) qRT-PCR of AR targets *PSA, NKX3-1*, and *SLC45A3* in VCaP cells expressing the noted shRNA constructs. Expression levels were normalized as described in Materials and methods; bars represent $\pm$ SEM of three biological replicates, each with three technical repeats. (C) qRT-PCR of *PSA* and *NKX3-1* from VCaP cells expressing the noted shRNA constructs alongside cDNAs expressing vector control (Vector), wild-type (WT) PRMT5, or a catalytically dead PRMT5 mutant (G365A/R368A). Data and error bars represented as in (B). (D) Top panels: cartoons of the *PSA* and *NKX3-1* loci. ERG and AR binding sites (and control regions) are noted and numbered relative to the transcription start site (TSS) as described in Materials and methods. Bottom panels: ERG, PRMT5, and AR ChIP qPCR for the noted regions of *PSA* (left) or *NKX3-1* (right) in VCaP cells upon ERG or PRMT5 knockdown. Normalization to IgG control ChIP is as described in Materials and methods; error bars represent $\pm$ SEM of three biological replicates, each with three technical repeats. (E) Heatmap visualization of AR binding from ChIP-sequencing data as determined by normalized reads across the AR Cistrome (Materials and methods) in replicate samples induced using AR ligands (DHT or R881 as indicated) and harboring inducible PRMT5 shRNA1 (sh1), shRNA2 (sh2), or shRNA3 (sh3) compared to NTC shRNA. 1659 peaks show differential binding with at least 1.5 fold difference (p-value of 0.01, q-value 0.151). The majority of differentially bound sites exhibit increased binding (6% of the total Cistrome) under PRMT5 knockdown conditions.

The following figure supplements are available for figure 2:

**Figure supplement 1.** AR signature analysis.

**Figure supplement 2.** ERG and PRMT5 effects on AR target genes are specific.

**Figure supplement 3.** ERG, AR and PRMT5 recruitment.

attenuate *PSA, NKX3-1*, and *SLC45A3* expression (*Mounir et al., 2015*). Under these conditions, PRMT5 knockdown restored the expression of these genes to baseline levels, yet had no effect on their expression in the background of the DNA-binding defective (i.e. inactive) ERG mutant ('Dx', *Figure 2—figure supplement 2D*). These data indicate that PRMT5's ability to repress AR function is dependent on ERG.

We next used chromatin immunoprecipitation (ChIP) to investigate ERG, AR, and PRMT5 recruitment to the AR targets *PSA* and *NKX3-1* following modulation of ERG or PRMT5 expression. ERG knockdown in VCaP reduced its recruitment to previously characterized (*Wei et al., 2010*) binding sites on both genes (*Figure 2D*). We also observed PRMT5 recruitment to these same sites, which was dramatically reduced upon ERG knockdown. Conversely, ERG expression in 22Rv1 cells induced PRMT5 recruitment to these same sites (*Figure 2—figure supplement 3A*). In VCaP, PRMT5 knockdown reduced its own recruitment to both genes but had virtually no effect on ERG recruitment (*Figure 2D*). As expected (*Yu et al., 2010*), ERG expression in 22Rv1 reduced AR recruitment to both genes (*Figure 2—figure supplement 3A*), and ERG knockdown in VCaP increased AR recruitment (*Figure 2D*). Like ERG, PRMT5 knockdown in VCaP strongly induced AR recruitment (*Figure 2D*).

To extend these findings to a genome wide scale, we performed AR and ERG ChIP-seq experiments in androgen (DHT or R1881) stimulated VCaP cells upon PRMT5 knockdown (Materials and methods; available PRMT5 antibodies did not work in our hands for ChIP-seq). ERG and AR recruitment were robust in these experiments, as judged by recovery of the canonical DNA binding sites of these proteins (*Figure 2—figure supplement 3B–D*). In agreement with directed ChIP experiments, AR ChIP-seq demonstrated that PRMT5 knockdown increased the recruitment of AR at a subset of peaks (6% p-val 0.01, q-val 0.151), (*Figure 2E*), but did not significantly affect ERG binding on the same set of sites. Collectively, these data support a model where ERG recruits PRMT5 to AR targets, and PRMT5 is required for ERG-dependent attenuation of AR binding to specific regulatory regions.

In many cell types, PRMT5 represses gene expression through symmetric di-methylation of histone H4 at arginine 3 (H4R3me2s) (*Yang and Bedford, 2013*), suggesting this as a mechanism for PRMT5 function at AR target genes. However, ERG expression in 22Rv1 cells did not alter H4R3me2s levels at the *PSA* or *NKX3-1* loci (*Figure 3—figure supplement 1*). We therefore hypothesized that PRMT5 directly methylates AR. To test this, we immunoprecipitated AR from ERG-positive VCaP cells, and blotted for AR, mono-methyl arginine (MMA), or symmetric di-methyl arginine (SDMA). MMA and SDMA signals were indeed detected in AR immunoprecipitates from VCaP cells, and were reduced following knockdown of either ERG or PRMT5 (*Figure 3A*). To confirm these

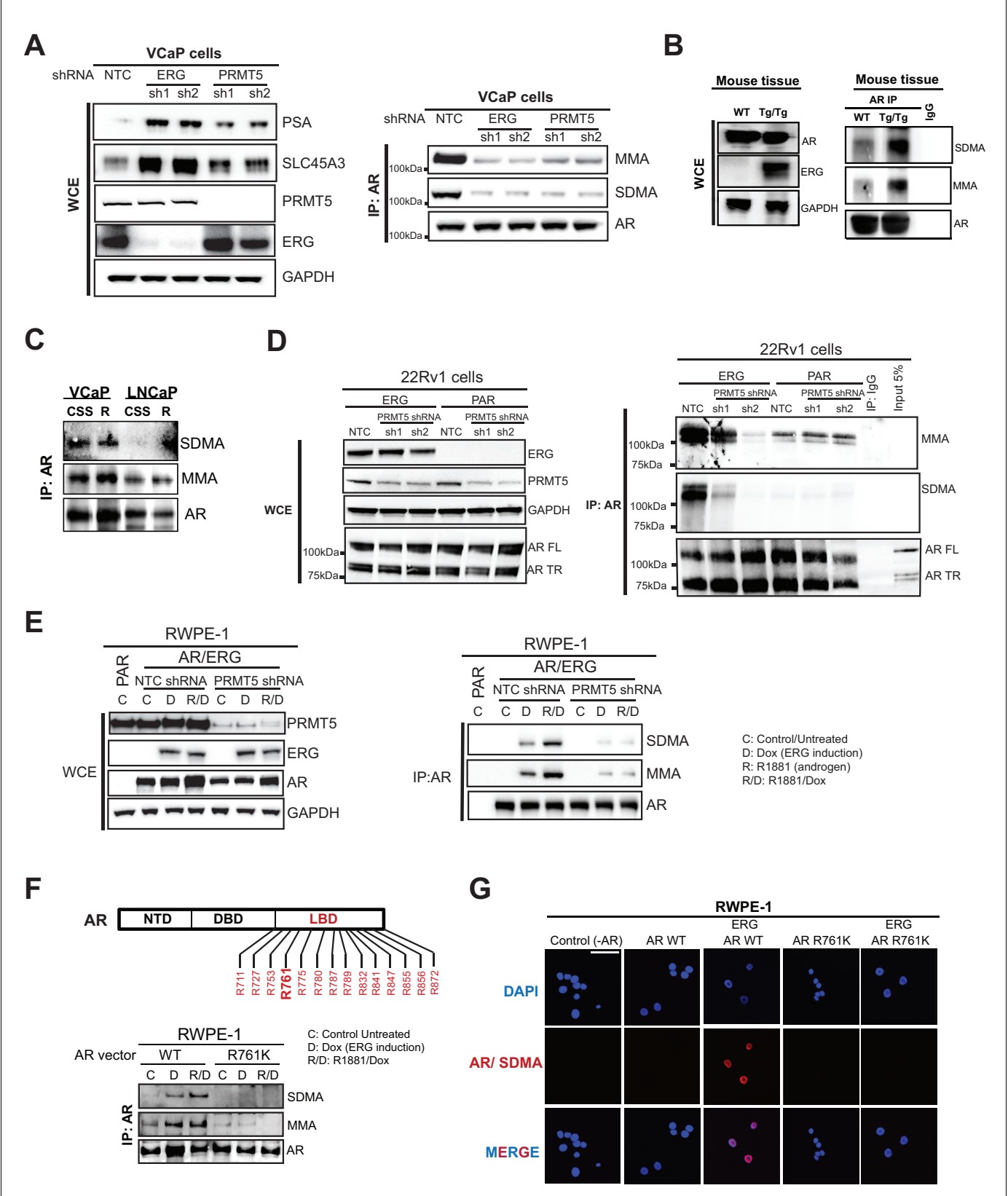

**Figure 3.** PRMT5 methylates AR on arginine 761. (A) (**A**) Left panel: western blots of noted proteins from VCaP whole cell extracts (WCE) after ERG or PRMT5 knockdown. Right panel: Western blot of AR immunoprecipitation (IP) from VCaP. SDMA: symmetric di-methyl arginine; MMA: mono-methyl

*Figure 3 continued on next page*

Figure 3 continued

arginine. (B) Left panel: Western blot analysis of noted proteins from homozygous *TMPRSS2:ERG* transgenic (Tg/Tg) and WT mouse tissues. Right panel: AR or IgG IP from mouse tissues followed by western blot analysis of MMA, SDMA and total AR levels. (C) Western blot of AR immunoprecipitation (IP) from VCaP and LNCaP cells grown in charcoal-stripped serum (CSS) and stimulated with 10nM R1881. SDMA: symmetric di-methyl arginine; MMA: mono-methyla arginine. (D) Left panel: western blot of noted proteins from 22Rv1 parental (PAR) or ERG-expressing (ERG) WCEs. FL: full-length; TR: truncated (lacking ligand binding domain, LBD). Right panel: Western blot of AR IP from 22Rv1. (E) Left panel: RWPE-1 parental (PAR) and AR and ERG-expressing (AR/ERG) cells targeted by PRMT5 knockdown (PRMT5 shRNA) or NTC shRNA were left either untreated (C) or treated with 100ng/ml doxycycline (D) in the absence or presence of 1nM R1881 (R) for 24 hr. Western blot analysis shows expression levels of PRMT5, ERG, AR and GAPDH from input samples (WCE). Right panel: Lysates were then used for AR immunoprecipitation (AR IP) followed by western blot analysis using antibodies against MMA, SDMA or total AR levels. (F) Top panel: location of all arginines (R) in the AR LBD. NTD: N-terminal domain; DBD: DNA binding domain. Right panel: western blot of AR IPs from RWPE-1 cells expressing ERG with wild-type AR (AR WT) or R761K mutant. C: control untreated; D: Dox-treated (ERG induction); R: R1881-treated. Bottom panel: western blot analysis of MMA, SDMA and total AR levels from AR IPs in RWPE-1 cells expressing ERG with either wild-type AR (AR WT) or R761K mutant. C: control untreated; D: Dox-treated (ERG induction); R: R1881-treated. (G) Representative immunofluorescence images of Dox- and R1881-treated RWPE-1 cells expressing ERG or AR as noted above each column. AR/SDMA: proximity ligation signals using antibodies detecting AR and SDMA (see Materials and methods). Scale bar, 50 μm. Data shown is a representative example of three biological replicates.

The following figure supplements are available for figure 3:

**Figure supplement 1.** H4R3me2s ChIP.

**Figure supplement 2.** PRMT5 methylates AR in vitro.

**Figure supplement 3.** AR and ERG expression in RWPE-1, and mutation of AR LBD.

findings in an additional ERG-dependent model, we also immunoprecipitated AR from the prostates of wild-type and *TMPRSS2:ERG* transgenic mice, the latter of which show ERG-dependent hyperpro-liferative phenotypes (*Mounir et al., 2015*). Like VCaP, immunoprecipitated AR from *TMPRSS2:ERG* mouse prostates showed high mono- (MMA) and symmetric di-methylation (SDMA) levels compared to wild-type controls (*Figure 3B*). On the other hand, LNCaP cells, which bear a translocation in the ETS factor *ETV1*, do not show any SDMA signal on AR, and show reduced levels of MMA versus VCaP (*Figure 3C*), suggesting that AR symmetric dimethylation is unique to ERG versus ETV1. We next immunoprecipitated AR from 22Rv1 cells, which express wild-type AR as well as a roughly 80 kDa truncated AR variant that lacks its ligand binding domain (LBD) (*Dehm et al., 2008*). In these cells, ERG expression increased MMA and SDMA signals on wild-type AR but not the truncated vari-ant, and these ERG-dependent signals were reduced upon PRMT5 knockdown (*Figure 3D*), suggest-ing the AR LBD is mono- and symmetrically di-methylated in an ERG- and PRMT5-dependent manner.

To further understand if the observed AR methylation was directly dependent on PRMT5, we per-formed biochemical assays using purified PRMT5 (complexed with its requisite binding partner MEP50/WDR77) and AR LBD. PRMT5 activity, as judged by production of SAH (the by-product of SAM-dependent substrate methylation), was observed in the presence of AR LBD as substrate, but not in the presence of ERG ETS DNA binding domain or pointed (PNT) domain. PRMT5 activity in the presence of AR LBD further increased with the addition of ERG ETS domain protein to the reac-tion, but not with PNT domain (*Figure 3—figure supplement 2A*). Unlike PRMT5, purified PRMT1 showed no activity in the presence of AR LBD (*Figure 3—figure supplement 2B*). PRMT5 activity on AR, in the absence or presence of ERG ETS domain, was reduced with the addition of the tool PRMT inhibitor AMI-1 (*Cheng et al., 2004*). PRMT5 dependent methylation of the AR-LBD was also observed in western blots, where increased signal for MMA (but not SDMA, likely due to low overall PRMT5 activity in vitro and the distributive nature of this enzyme [*Wang et al., 2014*]) was observed at the correct size for the AR-LBD protein in the presence of PRMT5. This activity was increased by ERG ETS domain, and inhibited by AMI-1 (*Figure 3—figure supplement 2C–D*). Together this data indicates that AR LBD is a substrate of the PRMT5 enzyme in vitro, and that the ERG ETS domain can facilitate greater PRMT5 activity on AR.

To identify the arginine methylation site(s) on AR, we cloned AR cDNA and mutated all arginines in the LBD to lysine (see Materials and methods). We expressed each construct in AR-negative, ERG-

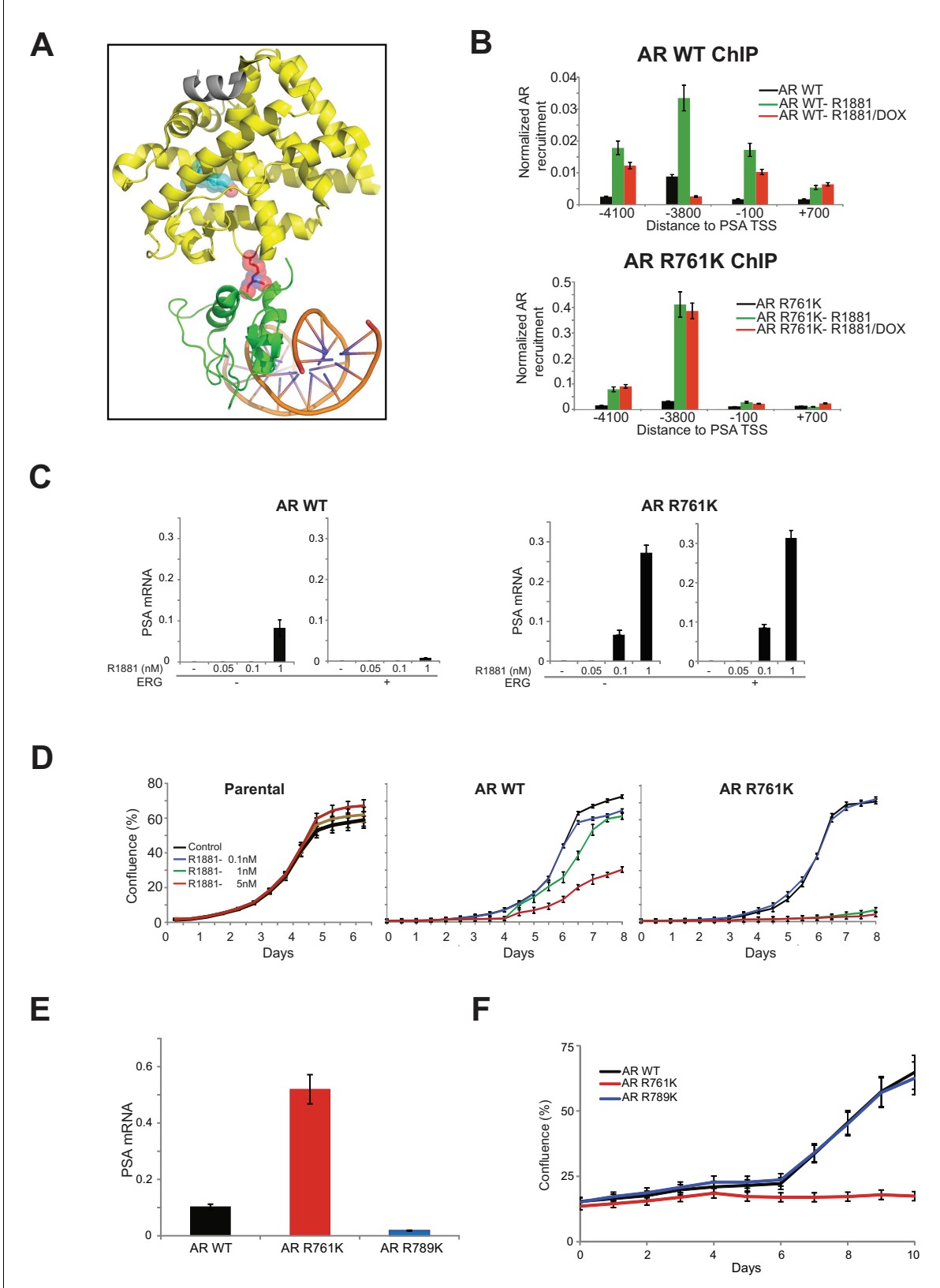

**Figure 4.** R761 methylation regulates AR recruitment, transcription, and proliferation. (**A**) Model of AR LBD (PDB: 2AO6; yellow) and AR DBD (PDB: 1R4I; green) interactions (see Materials and methods). A modeled di-methylated R761 is shown (red). Grey ribbon: TIF2 coactivator peptide. Cyan

*Figure 4 continued on next page*

*Figure 4 continued*

spheres: R1881. DNA is shown as orange/blue sticks. (**B**) AR ChIP qPCR for regions of the *PSA* gene as in *Figure 2D* from RWPE-1 cells expressing wild-type (WT, left) AR or AR R761K (right). DOX: ERG expression; R1881 is 1nM. Error bars represent ± SEM of three biological replicates, each with three technical repeats. (**C**) *PSA* qRT-PCR in RWPE-1 cells expressing WT AR (left) or AR R761K (right). Error bars represent ± SEM of three biological replicates, each with three technical repeats. (**D**) RWPE-1 parental cells and cells expressing either wild type AR (AR WT) or AR R761K mutant were left either untreated (control) or treated with 0.1, 1 or 5nM R1881 for 6 days and confluence measurements (see Materials and methods for description) were collected. Error bars represent ± SEM of three biological replicates, each with three technical repeats. (**E**) *PSA* qRT-PCR from VCaP cells expressing the noted AR constructs, grown in androgen-depleted media (charcoal-stripped serum). (**F**) VCaP cell proliferation upon expression of the noted AR constructs.

The following figure supplement is available for figure 4:

**Figure supplement 1.** AR R761K mutation effects in RWPE-1.

negative RWPE-1 prostate cells. R1881 stimulates exogenous AR to induce target gene expression in these cells. ERG expression represses this effect (*Figure 3—figure supplement 3A–B*) and induces PRMT5-dependent mono- and symmetric di-methylation of AR (*Figure 3E*). RWPE-1 cells co-expressing ERG with each of the AR constructs were assessed for SDMA and MMA modification of AR following immunoprecipitation. All mutants were expressed at equivalent levels to wild-type AR (*Figure 3—figure supplement 3C*). Only one AR mutant, R761K, completely lacked MMA and SDMA of AR in the presence of ERG (*Figure 3F*; *Figure 3—figure supplement 3D*). We confirmed these results in RWPE-1 cells using a proximity ligation assay to detect symmetric di-methylation of AR (Materials and methods). We detected strong proximity signals in RWPE-1 nuclei upon expression of wild-type AR and exposure to R1881 (which stimulates nuclear translocation of AR) that were dependent upon ERG expression. These signals were not detected in the AR R761K mutant cell line (*Figure 3G*). This demonstrates that arginine 761 is the likely target of ERG- and PRMT5-dependent AR methylation.

Available structural information (*Chandra V et al., 2008*; *Emsley et al., 2010*; *Helsen et al., 2012*) (see Materials and methods) suggests R761 methylation on AR may affect LBD interactions with the DNA binding domain (DBD, *Figure 4A*), resulting in altered DNA binding and gene activation. To test this, we evaluated the recruitment of WT versus R761K mutant AR in RWPE-1 cells treated with R1881, using the *PSA* locus as a model AR-regulated gene. Relative to WT AR, the AR R761K mutant showed enhanced recruitment to *PSA* and increased *PSA* expression. Moreover, R761K mutation prevented ERG-dependent attenuation of AR recruitment to *PSA* and *PSA* expression (*Figure 4B,C*). In contrast to these effects, the AR R761K mutation did not affect overall AR protein levels, the ability of AR to shuttle into the nucleus upon R1881 induction (*Figure 4—figure supplement 1*), or the levels of ERG or PRMT5 recruitment to *PSA* (*Figure 4—figure supplement 1C*). These results indicate R761 methylation mediated through ERG and PRMT5 attenuates ligand-dependent AR activation, likely through modulating interactions between the LBD and DBD.

Increased AR activity drives PC proliferation (*Chen et al., 2004*). However, we noted that RWPE-1 cells expressing AR R761K, despite increased AR activity, were more prone to R1881-induced growth arrest (*Figure 4D*). This suggested that heightened AR activity via R761K mutation (or loss of R761 methylation) may block proliferation in certain contexts. To further explore this, we expressed R761K mutant AR in VCaP cells. Despite transcriptional hyperactivity, as judged by increased *PSA* expression relative to either WT or an irrelevant R789K mutant AR, VCaP cells expressing AR R761K proliferated poorly in reduced-androgen media (*Figure 4E and F*).

Detailed mechanisms of how the tumor context affects AR and other lineage oncogenes to switch their function from lineage differentiation to proliferation in cancer has remained elusive (*Garraway and Sellers, 2006*). Our results indicate that a primary effect of ERG in facilitating PC proliferation is modulating AR function. We propose that ERG- and PRMT5-dependent methylation of R761 on AR reduces AR recruitment to genes that would otherwise induce differentiation, yet R761 methylation allows sufficient AR function to promote proliferation. While our results raise the question of how R761-methylated AR may still facilitate proliferation, R761 methylation could prove a relevant biomarker for AR-mediated proliferation versus arrest in *TMPRSS2:ERG* positive cells. Finally, as PRMT5 is an enzyme required downstream of ERG in facilitating AR proliferative function,

exploring therapeutic PRMT5 inhibition in *TMPRSS2:ERG* positive prostate cancers may be warranted.

## Materials and methods

### Cell Lines

VCaP, 22Rv1, LNCaP, PC3, RWPE-1, and 293T cells were obtained from ATCC and maintained in recommended media unless otherwise specified. Cell identities were verified by SNP analyses using ABI TaqMan SNP genotyping assays (Asuragen) and tested for mycoplasma contamination using the MycoAlert Mycoplasma Detection kit (Lonza).

### shRNA screen and data analysis

22Rv1 and VCaP cells were used for a screen using a custom shRNA library (Cellecta, Inc.) targeting transcriptional and epigenetic regulators similar to a previously reported library (*Hoffman et al., 2014*). shRNA library design and construction and viral packing were performed as previously described (*Hoffman et al., 2014*). To obtain an MOI of 0.3, the required volume of virus was determined using a 10 point dose curve ranging from 0 to 1ml of viral supernatant in the presence of 10 ug/ml polybrene. Infection efficiency was determined by the percentage of RFP positive cells measured by FACS analysis. Screens were run in duplicate. For the 22Rv1 screen, 14.4 million cells were plated 24 hr prior to infection in T-225 flasks. On the day of infection, the culture media was replaced with fresh media containing 10 ug/mL polybrene and sufficient virus was added for an MOI of 0.3. 24 hr after infection, the culture media was replaced with fresh media containing puromycin. 72 hr following puromycin addition, cells were trypsinized, and 14.4 million cells were plated into new flasks. For the VCaP screen, 23.7 million cells were plated 24 hr prior to infection in T-225 flasks. On the day of infection, the culture media was replaced with fresh media containing 10 ug/mL polybrene and sufficient virus was added for an MOI of 0.3. 24 hr after infection, the culture media was replaced with fresh media and cells were allowed to recover for 72 hr prior to puromycin selection. 5 days following puromycin addition, cells were trypsinized, and 23.7 million cells were plated into new flasks. At each passage, an aliquot of cells was used to measure transduction efficiency determined by measuring the% RFP positive cells and was typically > 90%. Cells were maintained in culture and split when confluence reached 90% and at each passage, 14.4 million cells (22Rv1 screen) and 23.7 million cells (VCaP screen) were passaged into new flasks, ensuring a representation of >1000 cells/shRNA in the library and the% RFP positive cells was measured to ensure stability of the transduced population over time. When the cells reached 5-population doublings, 40 million cells were harvested by centrifugation and stored at −20°C. Purification of genomic DNA and PCR for library production were performed as previously described (*Hoffman et al., 2014*). shRNA screen data analysis was performed as previously described (*Hoffman et al., 2014*). For gene based hit calling, the The Redundant siRNA Activity or RSA metric was used as described (*Hoffman et al., 2014*). Briefly, the RSA down p-value is the statistical score that models the probability of a gene 'hit' based on the collective activities of multiple shRNAs per gene. The RSA down p-value reports the statistically significant genes causing a loss in viability. All hits showing an RSA score >$10^{-5}$ in the 22Rv1 screen and <$10^{-5}$ in the VCaP screen (total of 32 genes) were used for further analysis (see *Supplementary file 1* for full list).

### ERG immunoprecipitation and mass spectrometry analysis

Nuclear extracts of VCaP cells (~600 million cells) were pre-cleared using agarose beads (Trueblot rabbit kit, eBioscience) and used for pulldown with either an ERG antibody (Santa Cruz#353) or an anti-rabbit IgG antibody. All antibodies were pre-coupled to beads (AminoLink Plus kit, Thermo Scientific), washed then used for pulldown. Immunoprecipitations were eluted at pH2.5 followed by TCA protein precipitation, alkylated with iodoacetamide, and separated on a NuPage 4–12% Bis-Tris gradient gel (Invitrogen). Complete gel lanes were excised using a LEAP 2DiD robot and in-gel digested with trypsin (Tecan Freedom EVO 20). Peptide sequencing for the resulting resulting 16 digest samples was performed by liquid chromatography-tandem mass spectrometry using an Eksigent 1D+ high-pressure liquid chromatography system coupled to a LTQ-Orbitrap XL mass spectrometer (Thermo Scientific). Peptide mass and fragmentation data were searched against a

combined forward-reverse IPI database (v3.55) using Mascot 2.2 (Matrix Science). Peptide and protein validation were done using Transproteomic pipeline v3.3sqall (Institute for Systems Biology; [http://tools.proteomecenter.org/software.php]) using a false positive threshold of <1% for protein identifications (See *Supplementary file 2* for full list; in red are the ERG interactors also identified as shRNA screen hits).

## shRNA knockdown

The Dox-inducible shRNA vector (pLKO-Tet-On) was previously described, as were the sequences of the nontargeting control and ERG shRNA inserts and stable cell line generation(*Mounir et al., 2015*). PRMT5 shRNA sequences are as follows:

PRMT5 shRNA#1:
AGGGACTGGAATACGCTAATTCTCGAGAATTAGCGTATTCCAGTCCCT
PRMT5 shRNA#2:
AGGGACTGGAATACGTTAATTGTTAATATTCATAGCAATTAGCGTATTCCAGTCCCTPRMT5 shRNA#3:
GCGGATAAAGTTGTATGTTGTGTTAATATTCATAGCACAGCATACAGCTTTATCCGC

All shRNA were expressed from a puromycin resistant vector. Lentiviral production and cell transduction was as previously described (*Mounir et al., 2015*).

## cDNA vectors

The Dox-inducible ERG constructs named ERG or ERG DNAx used for stable and inducible expression in 22Rv1 cells were generated as previously described (*Mounir et al., 2015*). The cDNA expression rescue constructs were cloned into a pRETRO retroviral vector under a CMV promoter and containing a neomycin-IRES-YFP selection cassette. The HA-tagged PRMT5 sequence was generated synthetically to resist knockdown by all PRMT5 shRNAs used in this study and was cloned into a Gateway compatible entry vector. In order to resist knockdown by all PRMT5 shRNA sequences, 6–7 silent mutations (in red below) were introduced into the HA-PRMT5 cDNA sequence to produce a shRNA resistant version (HA-scPRMT5). For shPRMT5-1: the AGGGACTGGAATACGCTAATT target sequence was converted to CGCGATTGGAACACGTTGATT (underlines denote altered bases).

For shPRMT5-2: the GCGGATAAAGCTGTATGCTGT target sequence was converted to CAGAATCAAGCTCTACGCCGT (underlines denote altered bases).

Full sequence of scPRMT5 below:

scPRMT5 sequence: ATGTACCCCTATGACGTGCCAGATTACGCCATGGCGGCGATGGCGGTCGGGGGTGCTGGTGGGAGCCGCGTGTCCAGCGGGAGGGACCTGAATTGCGTCCCCGAAATAGCTGACACACTAGGGGCTGTGGCCAAGCAGGGGTTTGATTTCCTCTGCATGCCTGTCTTCCATCCCAGGTTCAAGCGCGAGTTTATTCAGGAACCTGCTAAGAATCGGCCCGGTCCCCAGACACGATCAGACCTACTGCTGTCAGGACGCGATTGGAACACGTTGATTGTGGGAAAGCTTTCTCCATGGATTCGTCCAGACTCAAAAGTGGAGAAGATTCGCAGGAACTCCGAGGCGGCCATGTTACAGGAGCTGAATTTTGGTGCATATTTGGGTCTTCCAGCTTTCCTGCTGCCCCTTAATCAGGAAGATAACAC-CAACCTGGCCAGAGTTTTGACCAACCACATCCACACTGGCCATCACTCTTCCATGTTCTGGATGCGGGTACCCTTGGTGGCACCAGAGGACCTGAGAGATGATATAATTGAGAATGCACCAACTACACACACAGAGGAGTACAGTGGGGAGGAGAAACGTGGATGTGGTGGCACAACTTCCGGACTTTGTGTGACTATAGTAAGAGGATTGCAGTGGCTCTTGAAATTGGGGCTGATTTGCCCTCTAATCACGTCATTGATCGCTGGCTTGGGGAGCCCATCAAAGCAGCCATTCTCCCCACTAGCATTTTCCTGACCAATAAGAAGGGATTTCCTGTTCTTTCTAAGATGCACCAGAGGCTCATCTTCCGGCTCCTCAAGTTGGAGGTGCAGTTCATCATCACAGGCACCAACCACCACTCAGAGAAGGAGTTTTGTAGCTACCTGCAGTACCTGGAATACTTAAGCCAGAACCGTCCTCCACCTAATGCCTATGAACTCTTTGCCAAGGGCTATGAAGACTATCTGCAGTCCCCGCTTCAGCCACTGATGGACAATCTGGAATCTCAGACATATGAAGTGTTTGAAAAGGACCCCATCAAATACTCTCAGTACCAGCAGGCCATCTATAAATGTCTGCTAGACCGAGTACCAGAAGAGGAGAAGGATACCAATGTCCAGGTACTGATGGTGCTGGGAGCAGGACGGGGACCCCTGGTGAACGCTTCCCTGCGGGCAGCCAAGCAGGCC-GACCGCAGAATCAAGCTCTACGCCGTGGAGAAAAACCCAAATGCCGTGGTGACGCTAGAGAACTGGCAGTTTGAAGAATGGGGATCCCAGGTCACGGTAGTCAGCTCAGACATGAGGGAATGGGTGGCTCCAGAGAAAGCAGACATCATTGTCAGTGAGCTTCTGGGCTCATTTGCTGACAATGAATTGTCGCCTGAGTGCCTGGATGGAGCCCAGCACTTCCTAAAAGATGATGGTGTGAGCA

TCCCCGGGGAGTACACTTCCTTTCTGGCTCCCATCTCTTCCTCCAAGCTGTACAATGAGG
TCCGAGCCTGTAGGGAGAAGGACCGTGACCCTGAGGCCCAGTTTGAGATGCCTTATGTGG
TACGGCTGCACAACTTCCACCAGCTCTCTGCACCCCAGCCCTGTTTCACCTTCAGTCACCCTAA
TCGCGACCCCATGATTGACAACAACCGCTATTGCACCTTGGAATTTCCTGTGGAGGTGAACACAG
TACTACATGGCTTTGCCGGCTACTTTGAGACTGTGCTTTATCAGGACATCACTCTGAGTATCCG
TCCAGAGACTCACTCTCCTGGGATGTTCTCATGGTTTCCTATTCTGTTTCCCATCAAGCAGCCCA
TAACGGTACGTGAAGGCCAAACCATCGTGTGCGTTTCTGGCGATGCAGCAATTCCAAGAAGG
TGTGGTATGAGTGGGCTGTGACAGCACCAGTCTGTTCTGCTATTCATAACCCCACAGGCCGCTCA
TATACCATTGGCCTCTGA.

Generation of the PRMT5 catalytic dead mutant was performed by site-directed mutagenesis (QuickChange II, Agilent) through mutation of G365 to A and R368 to A (*Antonysamy et al., 2012*).

## Cell culture and proliferation assays

VCaP, 22Rv1, LNCaP and RWPE-1 cell lines (ATCC) were grown in vendor-recommended media and maintained in a humidified 5% $CO_2$ incubator at 37°C. Doxycycline (Dox, Sigma) was used at 100 ng/ml. Cell proliferation was measured in 6-well plates (Corning) using automated confluence readings (IncuCyte EX, Essen Bioscience). R1881 (Sigma) and charcoal-stripped serum (Omega Scientific) were used where indicated.

## Immunoprecipitation assays

2–5 ug of AR, ERG or IgG control antibody was coupled to 1mg of magnetic beads according to manufacturer's protocol (Invitrogen Dynabeads Antibody Coupling kit#143.11D). After coupling, 1 mg of the antibody/bead mixture was incubated with 1–5 mg of protein lysate overnight under rotation at 4°C. IP samples were then washed with RIPA buffer containing protease/phosphatase inhibitor cocktail for 3–4 washes and resuspended in non-reducing loading buffer, boiled and loaded on a gel for western blot analysis. Immunoprecipitations were performed using the following antibodies: anti-ERG antibody (Epitomics# 2805–1), anti-HA antibody (Roche#11815016001), anti-AR antibody (Thermo Scientific# MA5-13426) or anti-IgG antibody (Rockland Immunochemicals# RL011-0102).

## Western blot analysis

Procedures were previously described (*Mounir et al., 2015*) and the following antibodies were used at 1:1000 dilutions and incubated overnight at 4°C: ERG (Epitomics# 2805–1), PRMT5 (CST# 2252; SIGMA#P0493), GAPDH (Millipore#MAB374), H4 (CST#2592), AR (Santa Cruz#sc-7305), HA (Roche#11815016001), Symmetric Di-methyl arginine (SDMA, CST#13222), Mono-methyl arginine (MMA, CST#8711), TRIP12 (Abcam#ab86220), EIF4E (CST#9742), CDC42 (BD Transduction Laboratories#610929), HDAC1 (CST#2062), SMARCB1/SNF5 (Bethyl Laboratories# A301-087A), SMARCE1/BAF57 (Bethyl Laboratories# A300–810A).

## Microarray and pathway analyses

Generation of labeled cDNA, hybridization to Affymetrix U133plus2 human arrays, and data normalization were performed as described (*Mounir et al., 2015*). For the candidate signatures in Figure supplement 2A (*Malik et al., 2015*; *Mendiratta et al., 2009*; *Nelson et al., 2002*), a two-tailed fisher's exact test was used to determine if probesets representing genes in those signatures were under- or over-represented in the set of probesets that were up- or down-regulated at least 1.5-fold compared to expressed but non-differentially-expressed probesets, with a nominal p-value of 0.05 or less. For an unbiased approach (*Figure 2A*), pathways derived from GO terms and transcription-factor networks were analyzed for overrepresentation via a one-tailed interpolated fisher's exact test, using genes that varied 1.5-fold or more with a nominal p-value of 0.05 or less compared to all genes represented on the array; Benjamini-Hochberg (BH) correction was then applied to these p-values (*Wiederschain et al., 2007*). The VCaP microarray dataset (*Figure 2A*) is available at the NCBI Gene Expression Omnibus (accession number GSE65965).

## Signature correlation analysis

Black line (Figure supplement 2A) represents expressed probe set position and is ranked by average fold-change. Blue, green, and red lines indicate where the probe sets mapping to genes in the androgen receptor activation signatures appear in our data set and show the cumulative sum of the probe sets in the androgen receptor activation signatures that overlap with our gene list (only the highest expressing probe set was used per gene). The dashed line represents the hypothetical cumulative sum for a random list of genes that are unenriched.

## RNA isolation and qRT-PCR

RNA isolation was performed as previously described (Mounir et al., 2015). Taqman reactions (Applied Biosystems) were performed using Gene Expression master mix, FAM-labeled probes for PSA, NKX3-1 and SLC45A3 and VIC-labeled probe for Beta-2-macroglobulin (B2M) as a normalization control. Samples were run on a 7900HT Real-Time PCR machine (Applied Biosystems) and data was analyzed and normalized according to manufacturer's instructions ($2^{-\Delta Ct}$ method).

## Chromatin immunoprecipitation (ChIP)

VCaP, 22Rv1 and RWPE-1 cells were treated as specified followed by cross-linking with 1% formaldehyde for 10 min. Cells were next lysed in 1% SDS and sonicated until DNA ladder is below 1 kb (Diagenode). Sheared chromatin was then used for IP with specific primary antibodies (2-4 ug; previously tested) pre-complexed with Protein A/G Dynabeads and incubated overnight under rotation at 4°C. The next day, ChIP samples were washed with RIPA buffer and TE followed by reverse crosslinking using 1%SDS and 30 ug/ml proteinase K (Invitrogen) at 65°C for 6 hr with beads. The eluates were then purified using the QIAquick PCR purification kit (Qiagen) and used for qPCR with the following primer sets:

PSA -4100: acctgctcagcctttgtctc AND ttgtttactgtcaaggacaatcg
PSA -3800: agaattgcctcccaacactg AND cagtcgatcgggacctagaa
PSA -100: cttccacagctctgggtgt AND aaaccttcattccccaggac
PSA +700: agccccagactcttcattca AND atgcagatttggggaatcag
NKX3-1 -2800: gagagcagctgttcctccac AND acgagcctttccacctttc
NKX3-1 -200: agggaggagagctggagaag AND tcctccctaggggattcct
NKX3-1 +2150: accaggatgaggatgtcacc AND cagggacagagagagccttg
NKX3-1 -+10800: tctctcgttggctcctgatt AND ccagcttttgttccttcctg
NKX3-1 +62100: cggtttattgcccatgaaga AND aacagggctcacagtgcttt

## ChIP-seq

VCaP cells harboring Dox inducible shRNAs targeting PRMT5 where grown to 80% confluency and re-seeded (day 0) into full media containing 100 ng/ml Doxycycline. On day 3 media was replaced with 'hormone reduced' media, containing 100 ng/ml Doxycycline. Cells were stimulated with adding indicated ligands (DHT (Sigma), R1881 [Sigma]) or vehicle on day 4 and harvesting of cells was performed on day 5. Cells were harvested by fixation using 1% methanol free formaldedyde (Polysciences, Cat#18814) in PBS at room temperature. Fixation was stopped after 8 min by replacing fixation buffer with ice cold PBS containing 125 mM glycine and 5 mg/ml BSA. Cells were further washed once using ice-cold PBS and re-suspended into 500 ul of PBS containing Complete Protease Inhibitors (Roche). Cells where then pelleted and supernatant removed, and the resulting pellet either snap frozen in liquid Nitrogen or immediately re-suspended in lysis buffer for further processing.

For ChIP the resulting cell lysate was sonicated using a Covaris E210 instrument according to manufacturers recommendations. Each ChIP reaction was performed using soluble fraction chromatin corresponding to 7.5 ug purifed DNA and 4 ug of antibodies. Antibodies were allowed to bind overnight before capture on protein A magnetic beads (Invitrogen, Dynal). Bound beads where washed 4 times in RIPA buffer containing 500 mM LiCl, and 2 times with TE buffer before being re-suspended in containing 100 mM NaHCO$_3$ and 1% (w/v) SDS. Crosslink reversal was done at 65C°C for 6 hr and ChIP DNA were isolated using DNA purification beads (MagBio). ChIP-seq libraries were generated using the KAPA HTP library preparation kit (Kapa biosciences). All handling of

samples after sonication was done on using a Sciclone NGS Workstation (P/N SG3-31020-0300, Per-kinElmer). Sequencing was performed on an Illumina NextSeq500 instrument.

Reads passing Illumina standard QC were mapped to genome version Hg19 using BWA, and binding sites ('peaks') were identified using MACS2, evolutionary conservation scores at peak loca-tions was calculated using Phastcons and enriched DNA motifs using MDscan and Seqpos. These were performed using the ChiLin QC pipeline (liulab.dfci.harvard.edu/WEBSITE/software). Peaks from AR ChIP-sequencing samples with a MACS2 enrichment score higher than 10 were extended to a uniform 400 bp across all samples and overlapping peaks where collapsed to generate a union of all peaks. This resulted in a Cistrome of 25,593 peaks that were used in all genome wide ChIP analyses. Using the features of this Cistrome as GTF, read counts from BWA mapped Bam files were processed using the Qlucore 3.1.19 software. Heatmaps and statistical test (two-sided t-test using correction for multiple hypothesis testing) of differential binding scores on the 25,593 features were performed in Qlucore v3.1.19. The AR and ERG ChIPseq datasets are available at the NCBI Gene Expression Omnibus (Accession number GSE79128).

## Proximity ligation assay (PLA) and microscopy

RWPE-1 cells were treated as specified, fixed with 4% paraformaldehyde for 45 min at room temper-ature (Electron Microscopy Sciences), blocked with 5% goat serum, 0.5% Triton X-100 in PBS for 2 hr and incubated with 1:50 dilutions of AR (LSBIO #LS-C87494) and symmetric di-methyl arginine anti-bodies (CST#13222) in 5% goat serum and 0.05% Triton X-100. Fixed samples were incubated over-night at 4°C in primary antibody before incubations with proximity ligation assay (PLA) secondary antibodies (Duolink, Sigma).The secondary antibody incubation, ligation, amplification and final wash steps were performed according to the manufacturer's specifications. Confocal microscopy was per-formed using an LSM 510 META (Carl Zeiss, Inc.) with a 40x C-Apochromat objective, NA 1.2. Images were collected and processed using Zen software (Carl Zeiss, Inc.).

## AR mutagenesis and stable cell line generation

Full length *Homo sapiens* androgen receptor (AR) sequence (transcript variant 1; NM_000044.3) was synthesized to include a 5' NotI and 3' BamHI sites and used as a template for the mutagenesis of each arginine in the ligand binding domain of AR into lysine (QuikChange XL site-directed mutagen-esis kit, Agilent). Following sequence verification to ensure mutation incorporation, each AR mutant sequence was cloned into the pLVX vector via the 5' NotI and 3' BamHI as previously described (*Mounir et al., 2015*).

## Immunofluorescence and microscopy

Immunofluorescence of RWPE-1 cells was performed by fixing cells for 45 min at room temperature by adding 4% paraformaldehyde (Electron Microscopy Sciences) and incubated with 1:50 dilution of AR antibody (LSBIO #LS-C87494). Confocal microscopy was performed using an LSM 510 META (Carl Zeiss, Inc.) with a 40x C-Apochromat objective, NA 1.2. Images were collected and processed using Zen software (Carl Zeiss, Inc.).

## AR structural model

The heterodimeric structure of PPARγ-RXRα (PDB Code: 3DZY) was used as a template to overlay individual domain structures of AR including the DBD in complex with DNA (PDB Code: 1R4I) and the LBD in complex with coactivator peptide TIF2(iii) and ligand R1881 (PDB Code: 2AO6). A super-position was achieved using secondary structure matching in COOT (*Emsley et al., 2010*). The AR DBD was superposed onto chain A of RXRα and the AR-LBD was superposed onto chain B of PPARγ, resulting in the final overlay shown in *Figure 4A*. A dimethylated Arg was superposed onto R761 (NCBI Reference Sequence: NP_000035.2; some publications may refer to it as R760 when using the previous Reference Sequence number) within the AR LBD using least squares fit and matching only mainchain atoms, after which the non-methylated Arg was removed from the resulting model. While structure determination of AR containing its intact DBD and LBD has remained elusive, structural data of each individual domain, as well as intact structures within the nuclear receptor family, lead to a valuable understanding of interdomain communication. A previous study utilized the heterodimeric structure of PPARγ and RXRα to apply in silico three-dimensional alignment and docking analysis,

followed by mutational analysis, to propose a DBD-LBD interface within AR, including R761 (*Figure 4A*) (*Chandra V et al., 2008*; *Helsen et al., 2012*). We hypothesize that R761 is involved in key interactions at this interface and that its methylation would add hydrophobicity, eliminating any polar interactions, as well as steric bulk. This disruption at the DBD-LBD interface could result in destabilization of the quaternary structure, resulting in an inhibitory effect on AR activation. While AR R761K closely mimics the properties of wt AR, the substitution eliminates the possibility of methylation by PRMT5 and, therefore, eliminates the possibility of this type of disruption, accounting for the observed increase in activation.

### AR protein expression and purification

The gene encoding human AR LBD (residues 663–919), was inserted into a pGEX-6P-1 vector and expressed as a GST-tagged fusion protein in BL21 Star (DE3) cells. Cells were grown in TB2 medium containing 10 µM dihydrotestosterone (DHT) and induced with 1 mM IPTG for 14 –16 hr at 16°C. Cells were resuspended in buffer A containing 50 mM Tris-HCl (pH 7.3), 150 mM NaCl, 10% glycerol, 0.25 mM TCEP, and 10 µM DHT, to which 50 µg/ml DNase I and protease inhibitor cocktail (Roche) were added. Cells were lysed using an M-110L Microfluidizer at 18,000 psi, followed by the addition of 0.5% CHAPS to the lysate prior to high speed centrifugation. For one-step batch purification, the soluble extract was incubated with 2 ml of glutathione sepharose 4 fast flow medium (GE Healthcare) for 1 hr at 4°C with rotational mixing. The sepharose medium was washed in buffer A with the addition of 0.5% CHAPS. Elution was accomplished by resuspending and incubating the media for 10 min in the wash buffer plus 10–20 mM reduced glutathione. The eluted fractions were then combined and concentrated to 0.3 mg/ml.

### PRMT5 methyltransferase assays

PRMT5 enzymatic activity was assessed by monitoring *S*-adenosyl-L-homocysteine (SAH) product formation utilizing liquid chromatography-tandem mass spectrometry (LC-MS/MS). 0.5–1 uM of PRMT5/MEP50 recombinant enzyme (BPS, cat#51045) was incubated with 50 uM SAM, 2 uM GST-AR LBD and/or 5 uM ETS or PNT ERG protein for 2 hr at 37°C. Reactions were quenched to 0.1% HCOOH followed by addition of $[\beta,\beta,\gamma,\gamma-^2H_4]$-SAH (SAH-D$_4$) in 20% DMSO as an internal standard for MS quantification. Samples were sonicated with a Hendrix SM-100 sonicator (Microsonics Systems) and centrifuged. SAH was separated from the reaction mixture by reversed phase chromatography using polar endcapped C18 reversed phase columns (Synergi Hydro-RP, 2.5 µm, 100 Å, 20 x 2 mm, Phenomenex) and detected using a 4000 QTRAP Hybrid Triple Quadrupole/Linear Ion Trap LC-MS/MS system (AB Sciex).

## Acknowledgements

We thank Huili Zhai, Veronica Sanz-Vash, Victor Lin, Yi Yang, Alyson Freeman, Konstantinos Mavrakis, Kenneth Crawford, Jessica Cherry, Aron Jaffe, Margaret McLaughlin, Nanxin Li, Frank Stegmeier, Pascal Fortin, Gyorgy Petrovics, Shiv Srivastava, and Wojciech Wrona for helpful discussions.

## Additional information

#### Competing interests

ZM: was the recipient of presidential postdoctoral fellowship from the Novartis Institutes for Biomedical Research and is an employee of Genentech. FL, CAK, MSch, GM, GH, NR, YF, DRK, CQ, MF, RM, JD, TS, NK, WRS, RAP: employee of Novartis Institutes for Biomedical Research. AH: is an employee of Organovo. GB: is an employee of Celgene. MB: is a consultant to Novartis and the recipient of sponsored research support from Novartis. The other authors declare that no competing interests exist.

### Funding

| Funder | Grant reference number | Author |
| --- | --- | --- |
| Novartis | Dana Farber/Novartis drug discovery program | Thomas Westerling<br>Gilles Buchwalter<br>Myles Brown |

All authors except Thomas Westerling, Gilles Buchwalter, and Myles Brown received no external funding for this work.

### Author contributions

ZM, Conception and design, Acquisition of data, Analysis and interpretation of data, Drafting or revising the article; JMK, CAK, MB, RAP, Conception and design, Analysis and interpretation of data, Drafting or revising the article; TW, GM, GH, NR, AH, YF, GB, Conception and design, Acquisition of data, Analysis and interpretation of data; FL, MSch, Acquisition of data, Analysis and interpretation of data, Drafting or revising the article; DRK, CQ, MF, JB, Conceived, executed, and interpreted experiments to produce AR and ERG protein, and to use these proteins to assess the ability of PRMT5 to methylate AR; MSch, RM, JD, Acquisition of data, Analysis and interpretation of data; TS, NK, WRS, Conception and design, Analysis and interpretation of data

### Author ORCIDs

Myles Brown, http://orcid.org/0000-0002-8213-1658
Raymond A Pagliarini, http://orcid.org/0000-0001-8920-8198

## Additional files

### Supplementary files

• Supplementary file 1. shRNA screen hits selective to ERG-positive prostate cancer

• Supplementary file 2. List of candidate ERG protein interactors identified by Mass Spectrometry analysis of an ERG pulldown

• Supplementary file 3. PRMT5 activated pathways

### Major datasets

The following datasets were generated:

| Author(s) | Year | Dataset title | Dataset URL | Database, license, and accessibility information |
| --- | --- | --- | --- | --- |
| Zineb Mounir, Raymond A Pagliarini, Joshua M Korn | 2015 | Testing gene expression changes in VCaP cells following PRMT5 knockdown | http://www.ncbi.nlm.nih.gov/geo/query/acc.cgi?acc=GSE65965 | Publicly available at the NCBI Gene Expression Omnibus (Accession no: GSE65965). |
| Thomas Westerling, Zineb Mounir, Raymond A Pagliarini, Myles Brown | 2016 | AR and ERG ChIP-seq in presence or absence of PRMT5 | http://www.ncbi.nlm.nih.gov/geo/query/acc.cgi?token=qhyzcwmulfmdvkr&acc=GSE79128 | Publicly available at the NCBI Gene Expression Omnibus (Accession no: GSE79128). |

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
