## [Decision Letter]

Thank you for submitting your work entitled "ERG signaling in prostate cancer is driven through PRMT5-dependent methylation of the Androgen Receptor" for consideration by *eLife*. Your article has been reviewed by two peer reviewers, and the evaluation has been overseen by a Reviewing Editor and Kevin Struhl as the Senior Editor.

The reviewers have discussed the reviews with one another and the Reviewing Editor has drafted this decision to help you prepare a revised submission. As you will see below, the reviewers have some concerns that will require quite an additional effort to address. Ordinarily, we allow just two months to return a revised version but we are concerned that this required experiments may take longer to complete. To help us assess the likelihood of a successful completion of the required additional work, please send us a note indicating how you plan to proceed and an estimate of the time it will take to complete the work.

Summary:

In this manuscript, the authors present two major findings: 1) PRMT5 methylates AR at R761 and these methylation results in decreased AR recruitment at target genes and decreased AR function and 2) ERG targets AR to be methylated. Overall, the findings address long-standing questions of ERG function in prostate tumorigenesis and possibly posit new therapies to ERG translocated prostate cancers. This would be certainly relevant to the prostate field. In addition, it is appropriate for a general audience for implications of a general enzyme that affects specific programs. The experiments are well done and logical. The manuscript is well written. However, at this point, there are several critical experiments that should be performed to solidify the conclusion.

Essential revisions:

1) Definition of the ERG and AR binding genome wide with and without PRMT5 and ERG knockdown and correlation with the gene expression is very important for the conclusion. Is AR binding specifically depleted in ERG co-bound sites? The changes genome wide may not reflect the selected sites, as highlighted by the effect of FOXA1 on AR. This does require time and effort. But AR and ERG ChIP-seq is well optimized and ChIP-seq today is cost effective so this is eminently doable.

2) One weakness of the manuscript, intrinsic to prostate cancer research, is the use of a single ERG positive cell line. So any hit of the screen could be ERG specific, or even more likely, VCAP specific. Since there are no other available ERG dependent cell lines, the authors should note in the manuscript this major caveat. It would be very helpful if the authors could examine xenografts or perform some validation of the mechanisms in primary human tissue samples. For example, authors can IP AR from several ERG positive and ERG negative prostatectomy specimens to assess for methylation.

3) Mass spec evidence of methylation at R761 would be helpful. Since you know what you're looking for, IP of AR in VCAP and another ERG negative line and looking for this should not be hard and would provide definitive evidence. The mutagenesis is highly suggestive, but could still change conformation in a way such that R761 is not the direct methylation site.

---

## [Author Response]

Essential revisions:

1) Definition of the ERG and AR binding genome wide with and without PRMT5 and ERG knockdown and correlation with the gene expression is very important for the conclusion. Is AR binding specifically depleted in ERG co-bound sites? The changes genome wide may not reflect the selected sites, as highlighted by the effect of FOXA1 on AR. This does require time and effort. But AR and ERG ChIP-seq is well optimized and ChIP-seq today is cost effective so this is eminently doable.

We thank the reviewers for raising this important point. To extend our findings beyond individual genes, we performed ChIPseq for both ERG and AR in the presence of the androgen molecules R1881 and DHT, and in the absence or presence of PRMT5 knockdown. This data is provided in Figure 2 and Figure 2—figure supplement 3; raw data is deposited at the NCBI Gene Expression Omnibus under accession number GSE79128. In agreement with our data from directed analyses of the AR targets *KLK3* (aka *PSA*) and *NKX3-1*, the ChIPseq dataset shows a significant increase of AR recruitment upon PRMT5 knockdown at 6% of the total AR-bound cistrome. Genome-wide recruitment of ERG was not significantly affected upon PRMT5 knockdown, including to the sites where AR recruitment was increased (this data is deposited in NCBI, but no figures added to the paper due to the lack of significant findings). This data agrees with our model that ERG recruits PRMT5 to attenuate AR activity to a significant subset of its target genes.

2) One weakness of the manuscript, intrinsic to prostate cancer research is the use of a single ERG positive cell line. So any hit of the screen could be ERG specific, or even more likely, VCAP specific. Since there are no other available ERG dependent cell lines, the authors should note in the manuscript this major caveat. It would be very helpful if the authors could examine xenografts or perform some validation of the mechanisms in primary human tissue samples. For example, authors can IP AR from several ERG positive and ERG negative prostatectomy specimens to assess for methylation.

We thank the reviewers for acknowledging the challenges of obtaining appropriate prostate models and patient specimens. To address the general lack of *TMPRSS2:ERG* positive models, we have explored the ERG/PRMT5 interaction using additional model systems. First, in Figure 2 we expressed exogenous ERG, or a DNA-binding defective mutant of ERG (DNA binding defective, or 'Dx', which is mutant for two key arginines that contact DNA), in AR-positive 22Rv1 prostate cells. This demonstrates that the ERG/PRMT5 interaction can be observed outside VCaP. Furthermore, PRMT5 shRNA in this line demonstrates that the band observed in IP experiments is indeed PRMT5, and further suggests that DNA binding is not required to observe this interaction. Second, in Figure 1—figure supplement 1, we perform reciprocal PRMT5 and ERG IP’s in AR-negative 293 and PC3 cells, respectively. These data show that the ERG/PRMT5 interaction can occur in the absence of AR. Third, in Figure 1—figure supplement 1, we generated truncated mutants of ERG, and expressed these constructs in 293 cells. All of these constructs contain the ETS DNA binding domain, suggesting that this domain is required for PRMT5 interaction. Unfortunately, constructs deleted for the ETS DNA binding domain could not be expressed to further confirm this finding. Fourth, in Figure 3, we IP’ed AR from prostate tissues of *TMPRSS2:ERG* transgenic mice vs. control mice (Mounir et al., Oncogene 2015); AR immunoprecipitates from the *TMPRSS2:ERG* mouse prostates show increased SDMA and MMA signals at the correct size for AR. Fifth, we show in Figure 2—figure supplement 2 that the effects of PRMT5 knockdown are specific to ERG-dependent AR targets in VCaP, and that PRMT5 knockdown does not affect previously described AR-independent targets of ERG. Together, we feel this data strongly suggests the ERG/PRMT5 interaction, as well as increased AR arginine methylation in the presence of ERG, is present outside of the VCaP model, and that the ability of PRMT5 to mediate ERG activity is specific to AR-dependent genes.

We further attempted to confirm our findings in human tissue samples. In the absence of an AR-R761me2s-specific antibody for immunoblotting or IHC, we planned to use a chromogenic in situ proximity ligation assay (PLA) (Duolink In Situ – Brightfield, Sigma – Aldrich) with primary antibodies recognizing AR and symmetric di-methyl arginine (SDMA) to detect AR-R761me2s in human prostate cancer samples. First, we focused on optimizing PLA for formalin-fixed paraffin-embedded (FFPE) tissues, using FFPE RWPE-1 prostate cells, FFPE human prostate cancer samples, and two primary antibodies against Ki67 (Ki67 mouse monoclonal antibody, Dako; and Ki67 rabbit monoclonal antibody, Vector). While numerous individual spots representing positive signal were detected in the nuclei of RWPE-1 prostate cells, few, if any, spots were seen in the human tissues (see Figure 5), suggesting that pre-analytic variables, such as formalin-fixation time and storage conditions, may be have decreased the sensitivity of PLA to a point where the positive signal is below the limit of detection of the assay. Therefore, unfortunately, we do not currently have a robust means of detecting AR-R761me2s in human prostate cancer samples. The generation of a specific AR-R761me2s antibody would be of great benefit; however, we have not yet been able to generate such reagents.10.7554/eLife.13964.018Author Response Image 1.**DOI:**
http://dx.doi.org/10.7554/eLife.13964.018

3) Mass spec evidence of methylation at R761 would be helpful. Since you know what you're looking for, IP of AR in VCAP and another ERG negative line and looking for this should not be hard and would provide definitive evidence. The mutagenesis is highly suggestive, but could still change conformation in a way such that R761 is not the direct methylation site.

We agree that mass spectrometry data would be an important support of our findings. We made several attempts at mass spec identification of the arginine site using AR immunoprecipitated from VCaP cells. Unfortunately, we could not identify this site by mass spectrometry. In our hands, arginine methylation has been generally difficult to detect, including on histones, which are 'classic' PRMT5 targets. We suspect that R761me2s does not occur on the entire population of AR within the cell lysate, which 'masks' the detection of methylated AR by mass spec. This may be in line with the observation that PRMT5 knockdown does not affect AR recruitment to all possible sites in our ChIPseq experiments.

While not directly addressing the reviewers’ question, we provide additional supporting data that PRMT5 can methylate purified AR ligand binding domain in vitro. These biochemical assays demonstrate PRMT5 activity (SAH production, identification of methylation by western blot) on purified AR but not ERG, and that addition of purified ERG DNA binding domain can facilitate greater PRMT5 activity on AR. This data is provided in Figure 3—figure supplement 2.